# Diffuse Optical Monitoring of Cerebral Hemodynamics and Oxygen Metabolism during and after Cardiopulmonary Bypass: Hematocrit Correction and Neurological Vulnerability

**DOI:** 10.3390/metabo13111153

**Published:** 2023-11-16

**Authors:** Emilie J. Benson, Danielle I. Aronowitz, Rodrigo M. Forti, Alec Lafontant, Nicolina R. Ranieri, Jonathan P. Starr, Richard W. Melchior, Alistair Lewis, Jharna Jahnavi, Jake Breimann, Bohyun Yun, Gerard H. Laurent, Jennifer M. Lynch, Brian R. White, J. William Gaynor, Daniel J. Licht, Arjun G. Yodh, Todd J. Kilbaugh, Constantine D. Mavroudis, Wesley B. Baker, Tiffany S. Ko

**Affiliations:** 1Department of Physics & Astronomy, University of Pennsylvania, Philadelphia, PA 19104, USA; embenson@sas.upenn.edu (E.J.B.); yodh@physics.upenn.edu (A.G.Y.); 2Division of Neurology, Children’s Hospital of Philadelphia, Philadelphia, PA 19104, USA; menezesfor@chop.edu (R.M.F.); allafontant@gmail.com (A.L.); ranierin@chop.edu (N.R.R.); jharnajj@gmail.com (J.J.); breimannj@chop.edu (J.B.); bohyun.yun1@gmail.com (B.Y.); gerard_laurent@brown.edu (G.H.L.); licht@chop.edu (D.J.L.); bakerw@chop.edu (W.B.B.); 3Division of Cardiothoracic Surgery, Children’s Hospital of Philadelphia, Philadelphia, PA 19104, USA; aronowitzd@chop.edu (D.I.A.); gaynor@chop.edu (J.W.G.); mavroudisc@chop.edu (C.D.M.); 4Department of Anesthesiology and Critical Care Medicine, Children’s Hospital of Philadelphia, Philadelphia, PA 19104, USA; starrjp@chop.edu (J.P.S.); kilbaugh@chop.edu (T.J.K.); 5Department of Perfusion Services, Cardiac Center, Children’s Hospital of Philadelphia, Philadelphia, PA 19104, USA; melchiorr@chop.edu; 6Department of Chemistry, University of Pennsylvania, Philadelphia, PA 19104, USA; 7Division of Cardiothoracic Anesthesiology, Children’s Hospital of Philadelphia, Philadelphia, PA 19104, USA; lynchj3@chop.edu; 8Division of Cardiology, Children’s Hospital of Philadelphia, Philadelphia, PA 19104, USA

**Keywords:** hemoconcentration, cardiopulmonary bypass, cerebral hemodynamics, mild hypothermia, diffuse optics

## Abstract

Cardiopulmonary bypass (CPB) provides cerebral oxygenation and blood flow (CBF) during neonatal congenital heart surgery, but the impacts of CPB on brain oxygen supply and metabolic demands are generally unknown. To elucidate this physiology, we used diffuse correlation spectroscopy and frequency-domain diffuse optical spectroscopy to continuously measure CBF, oxygen extraction fraction (OEF), and oxygen metabolism (CMRO_2_) in 27 neonatal swine before, during, and up to 24 h after CPB. Concurrently, we sampled cerebral microdialysis biomarkers of metabolic distress (lactate–pyruvate ratio) and injury (glycerol). We applied a novel theoretical approach to correct for hematocrit variation during optical quantification of CBF in vivo. Without correction, a mean (95% CI) +53% (42, 63) increase in hematocrit resulted in a physiologically improbable +58% (27, 90) increase in CMRO_2_ relative to baseline at CPB initiation; following correction, CMRO_2_ did not differ from baseline at this timepoint. After CPB initiation, OEF increased but CBF and CMRO_2_ decreased with CPB time; these temporal trends persisted for 0–8 h following CPB and coincided with a 48% (7, 90) elevation of glycerol. The temporal trends and glycerol elevation resolved by 8–24 h. The hematocrit correction improved quantification of cerebral physiologic trends that precede and coincide with neurological injury following CPB.

## 1. Introduction

Despite decreasing mortality, neurodevelopmental deficits remain a common long-term outcome in children with severe congenital heart defects who require cardiac surgical interventions in the first weeks of life [1,2,3,4]. During such surgeries, cardiopulmonary bypass (CPB) is critical. CPB supplies exogenous blood flow and oxygen needed for patient survival, but the duration of CPB has been associated with increased post-operative white matter injury and neurodevelopmental delays at school age [5,6,7]. To date, both insufficient and excessive oxygen delivery relative to oxygen demand during CPB have been identified as potential risk factors for neurological injury [1,8,9], and dynamic changes in cerebral physiology and neurological injury biomarkers have also been observed over the first 24 h after bypass [10,11,12].

Clinical tools that continuously and concurrently monitor cerebral oxygen delivery and oxygen demand are needed to address these issues. The resulting information could enable physicians to identify periods of neurological risk due to mismatch of oxygen delivery and demand as well as to guide timely interventions during these vulnerable periods. Commercially available continuous-wave (CW) cerebral oximeters take steps towards this goal with emerging evidence of utility during CPB for the detection of acute cerebral desaturations [13,14]. However, CW systems show limited reliability and quantification accuracy at low oxygenation levels (such as those in children with cyanotic congenital heart disease) [15,16]. Furthermore, monitoring solely cerebral oxygenation does not permit users to determine whether cerebral oxygenation changes are caused by alterations in delivery or demand [15,16,17,18,19,20]. Concurrent use of transcranial doppler ultrasound (TCD) permits additional monitoring of cerebral perfusion, which has enabled rapid detection of cerebral emboli during CPB [21]. Nevertheless, TCD quantification is limited to macrovascular flow dynamics, requires a specialized operator, and faces challenges related to reproducibility, particularly in the context of longitudinal monitoring, due to inter-operator variability.

To address these limitations, here we combine advanced optical neuromonitoring techniques within a single device to continuously quantify microvascular changes in cerebral oxygen demand and delivery in a neonatal swine model of CPB. Specifically, we concurrently deploy non-invasive diffuse correlation spectroscopy (DCS) to measure cerebral blood flow (*CBF*), and frequency-domain diffuse optical spectroscopy (FD-DOS) to measure cerebral oxygen extraction fraction (*OEF*); concurrent measurement of *CBF* and *OEF*, in turn, provides continuous quantitative information about cerebral metabolic rate of oxygen (*CMRO*_2_). These techniques have been validated in infants and neonatal swine models [22,23,24,25,26], and their feasibility for clinical monitoring during CPB has been established [27,28,29,30,31,32]. However, the accuracy of existing in vivo data is limited by the assumption that hematocrit, defined as the ratio of the volume of red blood cells to the total volume of blood, remains constant. This study seeks to further advance translation of these techniques for CPB by critically examining and correcting for the hematocrit assumption in a reproducible preclinical swine model.

The present preclinical investigation undertakes several new directions. First, because profound changes in hematocrit can occur during initiation of CPB, we propose a novel theoretical model to incorporate hematocrit variation into diffuse optical approaches for *CBF* and *CMRO*_2_ quantification. Next, we apply and assess the impact of this hematocrit correction approach on *CBF* and *CMRO*_2_ quantification in vivo in a neonatal swine model of CPB. Continuous diffuse optical monitoring before CPB to up to 24 h after CPB is conducted for the first time to longitudinally characterize physiologic changes versus pre-CPB baseline. Further, we carry out concurrent invasive cerebral microdialysis sampling of biomarkers of bioenergetic dysfunction and neurological injury to characterize the timing of neurological vulnerability, provide cross-validation of physiologic changes measured by diffuse optical techniques, and provide hypothesis-generating data regarding underlying mechanisms of neurological vulnerability during and after CPB.

## 2. Materials and Methods

Non-invasive FD-DOS/DCS monitoring and invasive cerebral microdialysis sampling were conducted in one-week-old, female Yorkshire swine. The subjects were placed on cardiopulmonary bypass for three hours. Animals were randomized to different survival durations after decannulation. Survival duration was either zero, eight, twelve, eighteen, or twenty-four hours (these timepoints were selected to examine temporal changes in mitochondrial respiration and histopathological outcomes in the brain following CPB; results of these distinct analyses are reported elsewhere [33]). An overview of the procedure timeline can be seen in Figure 1. All animal care and procedures were approved by our Institutional Animal Care and Use Committee in accordance with the National Institutes of Health Guide for the Care and Use of Laboratory Animals.

### 2.1. Experimental Methods

#### 2.1.1. Induction and Sedation

Subjects were anesthetized with intramuscular injection of ketamine (30 mg/kg) and inhaled isoflurane (titrated between 1–2.5% to ensure absence of withdrawal response to toe pinch) before intubation. After intubation, ventilation was maintained by volume control at a tidal volume of 10–12 mL/kg, a positive end-expiratory pressure of 5 cmH_2_O, a fraction of inspired oxygen (FiO_2_) of 21%, and a respiratory rate titrated to an end-tidal CO_2_ of 38–42 mmHg. Venous and arterial access were established in the right femoral vein and artery, allowing for intravenous anesthetic infusions of fentanyl (25–200 μg/kg/min) and dexmedetomidine (0.5–2 μg/kg/min). These were titrated such that the piglet had no withdrawal response to toe pinch with 1–1.5% inhaled isoflurane.

#### 2.1.2. Monitoring Placement and Baseline Acquisition

After anesthetic and respiratory stabilization, physiologic monitoring was placed. Neuromonitoring consisted of a non-invasive diffuse optical probe and an invasive microdialysis catheter. The optical probe was sutured onto the left forehead, lateral to the midline, posterior to the snout, and anterior to the crown, enabling continuous FD-DOS and DCS measurements. The cerebral microdialysis probe was inserted into the right frontal cortex (CMA 71 Elite, mDialysis, Stockholm, Sweden) at a depth of about 1–1.5 cm into the brain parenchyma. The femoral arteries and veins were used for continuous arterial pressure monitoring and arterial and venous blood gas sampling. After placement of all monitoring devices, physiologic data were continuously acquired for a “baseline” period of five minutes during which changes in medication, ventilation, and physical manipulations were halted. To reduce heterogeneity across subjects, MAP was maintained between 45–65 mmHg via intravenous administration of vasodilators (milrinone, nitroglycerin, nicardipine) or vasopressors (epinephrine, phenylephrine).

#### 2.1.3. Initiation of Cardiopulmonary Bypass

The CPB circuit was primed with a combination of Plasma-Lyte A, fresh whole blood from donor swine, 500 IU heparin, 2 mEq/kg sodium bicarbonate, 1 mg/kg furosemide, and 450 mg calcium gluconate. To ensure that subsequent hematocrit after initiation of bypass would be >28%, the subject’s initial arterial blood hematocrit level was measured, and a weight-based estimate of red cell volume deficit was factored into determining the appropriate prime volume hematocrit needed to achieve bypass hematocrit goals [34]. The prime was typically hemoconcentrated to a volume of 200 mL with a hematocrit of ~35%; with these conditions, when the subject’s blood volume and the prime were combined on bypass, a hematocrit >28% was achieved. Following hemoconcentration, the prime volume was recirculated and sweep gas applied to achieve a pCO_2_ of 34–45 mmHg in the blood.

After preparation of the CPB prime and completion of baseline measurements, a heparin bolus of 4000 IU was given to the subject to prevent clot formation during cannulation. Cannulation was initiated after the activated clotting time (ACT) measured from an arterial blood sample was >450 s (i.e., in alignment with institutional practice). Cervical cannulation was performed with placement of an 8 Fr arterial cannula in the right carotid artery and a 10 or 12 Fr venous cannula in the right external jugular vein. Following cannulation, CPB was initiated, inhaled isoflurane administration was ceased, and 1% isoflurane was given through the CPB circuit. Fentanyl and dexmedetomidine infusions were continued and titrated to maintain adequate anesthetic depth. Throughout the three-hour duration of CPB, flow rates were kept above 100 mL/kg/min, MAP was maintained between 45–65 mmHg, heparin was administered to maintain ACT > 450 s, CPB FiO_2_ was titrated to achieve a partial pressure of arterial oxygen (*PaO*_2_) > 250 mmHg, and hemoconcentration and donor blood were used to maintain hematocrit >28%. Mechanical ventilation was maintained throughout CPB at rest settings (respiratory rate of 10 breaths per minute) to prevent atelectasis.

Immediately after bypass initiation, subjects were cooled to mild hypothermia (34 °C) at a rate no greater than 1 °C/min over a 20 min period. Subjects were maintained on mild hypothermic CPB (MH-CPB) for 140 min. Twenty minutes before the end of the three-hour CPB period, subjects were rewarmed to normothermia (37 °C) at a rate no greater than 1 °C/min.

#### 2.1.4. Decannulation and Post-Operative Survival

After three hours of CPB, the flow rate was reduced to zero, and animals were decannulated. Immediately after CPB cessation, protamine was administered at 1 mg per 100 IU heparin given throughout the CPB period to reverse the anticoagulatory effects of heparin. The right carotid and jugular vessels were ligated, and ventilator settings were returned to pre-CPB values. Fentanyl and dexmedetomidine infusions continued throughout the survival period. Isoflurane was not immediately used after CPB; however, it was restarted if there was a response to toe pinch. Animals continued to receive mechanical ventilation, intravenous sedation, and vasoactive support to sustain a MAP of 45–65 mmHg and an end-tidal CO_2_ between 38–42 mmHg.

After decannulation, animals were either immediately sacrificed or were monitored for eight, twelve, eighteen, or twenty-four hours prior to euthanasia via a bolus injection of potassium chloride.

### 2.2. Data Acquisition

#### 2.2.1. Diffuse Optical Neuromonitoring

*Frequency-domain diffuse optical spectroscopy (FD-DOS).* A customized, commercial FD-DOS instrument (Imagent, ISS Inc., Champaign, IL, USA) was used to continuously measure optical properties of cortical brain tissue. Twelve intensity-modulated (110 MHz) diode laser sources (four 690 nm, four 725 nm, and four 830 nm) and one photomultiplier tube detector housed in the instrument were coupled via optical fibers to the head probe (with source–detector separations of 0.7, 1.2, 1.7, and 2.2 cm). Selection of source–detector separation was informed by posthumous measurement of skull and scalp thicknesses which, in combination, typically ranged from 0.4–0.6 cm. Absolute absorption and reduced scattering coefficients were calculated using multi-distance fits of the collected AC intensity and phase data [23]. Cerebral oxy- and deoxy-hemoglobin concentrations ([*HbO*_2_] and [*Hb*], respectively, in μmol/L) were quantified from the multispectral tissue absorption measurements using chromophore extinction coefficients, as previously described (with an assumed cerebral tissue water volume fraction of 75%) [23]. This concentration information was used to derive total hemoglobin concentration (*THC* = [*HbO*_2_] *+* [*Hb*], μmol/L) and tissue oxygen saturation (*StO*_2_
*=* [*HbO*_2_]/*THC* × 100%). An index of cerebral blood volume (*CBV*) was derived from *THC* via:(1)$$CBV=\frac{{MW}_{Hb}\times THC}{\rho_{bt}\times [{Hgb}_{blood}]}=\frac{{MW}_{Hb}\times THC}{\rho_{bt}\times Hct\times MCHC},$$
where ${MW}_{Hb}$ = 64.5 kg/mol is the molecular weight of hemoglobin; $\rho_{bt}$ = 1.05 g/mL is the density of the brain tissue [35], and $\left[{Hgb}_{blood}\right]$ is the concentration of hemoglobin in blood (gdL^−1^). Note that $\left[{Hgb}_{blood}\right]$ was determined from measurements of hematocrit (Hct) in arterial blood samples via $Hct=\left[Hgb_{blood}\right]/MCHC$, where MCHC = 32 g/dL is the mean corpuscular hemoglobin concentration [36]. Finally, we fit the measured multispectral reduced scattering coefficients ($\mu_{s}^{\prime }(\lambda )$ at λ=690, 725, and 830 nm) to a Mie scattering model [37], i.e., $\mu_{s}^{\prime }(\lambda )=A{\left(\lambda /\left(500\;nm\right)\right)}^{-b}$, to derive the Mie scattering parameters *A* and *b.*

*Diffuse correlation spectroscopy (DCS).* DCS quantifies CBF by measuring fluctuations in light intensity. These fluctuations are primarily caused by the scatter of light off moving red blood cells; faster flow results in more rapid light fluctuations, which we quantified using the decay of the intensity temporal autocorrelation function (g_2_(τ)). DCS instrumentation was integrated into the same optical probe as the FD-DOS, thereby permitting co-located measurements as previously described [23,38]. DCS measurements were made using a source–detector separation of 1.7 cm. The DCS source was a continuous-wave, long-coherence, 785 nm laser (Custom iBeam smart WS, TOPTICA, Pittsford, NY, USA). Two single-mode detection fibers were coupled to single-photon-counting avalanche photodiodes (SPCM-AQ4C, Excelitas Technologies, Corp., Waltham, MA, USA). The g_2_(τ) curve was sampled at 20 Hz and averaged over 5 s (100 points), from which estimations of tissue blood flow (blood flow index, *BFI*) and the value of β in the Siegert relation were derived. Specifically, *BFI* and β were derived by numerically fitting the g_2_(τ) curve to the semi-infinite solution of the correlation diffusion equation [38]. The tissue absorption and reduced scattering coefficients at 785 nm estimated from concurrent FD-DOS measurements (i.e., via [*HbO*_2_], [*Hb*], and a Mie scattering model [37]) were inputs in this g_2_(τ) fit.

During analysis, large fluctuations in β due to measurement artifacts (e.g., motion and laser instabilities) were sometimes observed. These instabilities were filtered using an empirically developed method. For each subject, all timepoints were first fitted for both β and BFI. The fitted β from timepoints during the baseline period were averaged to determine a reference $\beta_{0}$. All timepoints where $\beta <(\beta_{0}-0.03)$ were removed. A beta difference greater than 0.03 was found to arise when there were obvious DCS measurement artifacts (which were often also apparent via manual inspection). After this filtering, the remaining timepoints were fit again for only the *BFI*, using a fixed $\beta =\beta_{0}$. The *BFI* obtained with this fixed β was used for all subsequent analyses.

#### 2.2.2. Cerebral Microdialysis

Microdialysis is a minimally invasive monitoring technique that directly samples metabolites from the interstitial fluid in the frontal cortex [39,40,41]. Sterile saline was infused at 1 µL/min for at least 30 min for equilibration before a baseline collection. Invasive cerebral microdialysis samples were collected at baseline (prior to CPB), in twenty-minute time intervals during CPB, and in sixty-minute time intervals during the survival period after decannulation (Figure 1). From these samples, the concentrations of pyruvate, lactate, glucose, and glycerol were measured using an automated ISCUS Flex™ Microdialysis Analyzer (CMA 71 Elite, mDialysis, Stockholm, Sweden), and data were processed using the associated mDialysis software (ICUpilot, Version 2.4.0.0). Data were excluded if values fell outside of the measurable range specified by the microdialysis analyzer. The lactate–pyruvate ratio (LPR) and glycerol were examined as key biomarkers of neurological injury. LPR is a strong predictor of anaerobic metabolism. Under normal conditions, LPR is expected to be <25, and LPR > 40 is a commonly used to indicate metabolic distress [39,42,43]. Glycerol levels are known to increase after neurological injury due to the degradation of glycerophospholipids during apoptosis [41,44,45]. Elevated glycerol levels have been shown to predict poor neurological outcomes [42,44,45]. Normal cerebral glycerol concentrations in humans have been reported in the range of 50–100 μM; glycerol concentrations exceeding 100 μM are associated with poor outcomes in TBI patients [46,47].

#### 2.2.3. Blood Gas

Systemic blood gas analysis is of relevance to the optical quantification of cerebral oxygen metabolism (discussed below). It provides accurate sampling of cerebral arterial blood-oxygen saturation (${SaO}_{2}$, %), hematocrit (Hct, %), and partial pressure of oxygen in the arterial blood (${PaO}_{2}$, mmHg). We collected femoral artery blood gas samples at baseline, once every hour during the bypass period, and once every three hours during the survival period (Figure 1). Femoral venous blood gas samples were also collected at the same timepoints. Immediate analysis of blood gas samples was completed using a blood gas analyzer (GEM 3000, Instrumentation Laboratory, Bedford, MA, USA).

### 2.3. Quantification of Cerebral Metabolic Rate of Oxygen (CMRO_2_)

Cerebral metabolic rate of oxygen (${CMRO}_{2}$) is the rate at which oxygen is metabolized by the brain. Quantification of ${CMRO}_{2}$ with optics offers a promising metric of metabolic distress. Assuming steady state conditions, we estimate *CMRO*_2_ via Fick’s law, i.e.,
(2)$${CMRO}_{2}={CaO}_{2}\times OEF\times CBF.$$

Here, ${CaO}_{2}$ is the arterial oxygen content (mL O_2_/dL blood), OEF is the cerebral oxygen extraction fraction, and CBF is cerebral blood flow [38]. The product of *CaO*_2_ and *CBF* is the rate of oxygen delivery to the brain. *OEF* is the fraction of oxygen in the arterial blood extracted by the tissue. In the next three subsections, we describe the measurements of *CaO*_2_, *OEF*, and *CBF* in detail.

We used Equation (2) as the basis to compute relative changes in *CMRO*_2_ from baseline ($rCMRO_{2}$):(3)$$rCMRO_{2}\equiv \frac{CMRO_{2}}{CMRO_{2,0}}=\frac{CaO_{2}}{CaO_{2,0}}\times \frac{OEF}{OEF_{0}}\times \frac{CBF}{CBF_{0}}$$

Here, the subscript “0” denotes the mean of the parameter across the baseline period (see Figure 1).

#### 2.3.1. Arterial Oxygen Content (*CaO*_2_)

Due to low O_2_ extraction from systemic arteries, a good approximation of *CaO*_2_ in the cerebral arteries is *CaO*_2_ in the femoral artery. We obtained *CaO*_2_ in the femoral artery from blood gas measurements of oxygen saturation (${SaO}_{2}$), hematocrit (Hct), and partial pressure of oxygen (${PaO}_{2}$) [48]:(4)$${CaO}_{2}=\left(1.39\frac{mL\;O_{2}}{g\;Hgb}\right)\times {SaO}_{2}\times \left(32.2\;\frac{g\;Hgb}{dL\;\;blood}\right)\times Hct+0.003\left(\frac{mL\;O_{2}}{dL\;blood\times mmHg}\right)\times {PaO}_{2}.$$

Here, the first term is the contribution from hemoglobin-bound oxygen, and the second term is the contribution from free dissolved oxygen. The latter contribution is small for normoxic conditions (<2%), but it is generally larger during hyperoxia (e.g., for the *PaO*_2_ target of 250 mmHg during CPB, the contribution is ~5%). Importantly, hematocrit is commonly assumed to remain constant over time, but this assumption is often violated in practice. For our experiment, 50% increases in Hct at the start of CPB were common. We thus used the full expression in Equation (4) to estimate *CaO*_2_ at each blood gas sample timepoint (see Figure 1).

#### 2.3.2. Oxygen Extraction Fraction (*OEF*)

The oxygen extracted by the tissue is the difference between *CaO*_2_ and cerebral venous oxygen content (*CvO*_2_). Thus, the *OEF* is defined by
(5)$$OEF\equiv \left({CaO}_{2}-{CvO}_{2}\right)/{CaO}_{2}\;.$$

If the free dissolved oxygen contribution to *CaO*_2_ is negligible (see Equation (4)), then *OEF* is well-approximated as $\left({SaO}_{2}-{SvO}_{2}\right)/{SaO}_{2}$, where ${SvO}_{2}$ is the cerebral venous blood-oxygen saturation. Under the hyperoxic conditions of this study, however, arterial free dissolved oxygen may not be negligible.

Accordingly, we directly used Equation (5) to estimate *OEF*. As discussed above, *CaO*_2_ was measured from arterial blood gas sampling. Measurement of *CvO*_2_ based on invasive cerebral venous blood sampling, however, was not feasible due to bleeding risks. Instead, we assumed that the free dissolved oxygen contribution is negligible for venous blood (this assumption is supported by our femoral venous blood gas sampling wherein partial pressure of oxygen never exceeded 100 mmHg and had an average of 33 mmHg during baseline and 47 mmHg during bypass). Thus, *CvO*_2_ was computed as:(6)$${CvO}_{2}=\left(1.39\frac{mL\;O_{2}}{g\;Hgb}\right)\times {SvO}_{2}\times \left(32.2\;\frac{g\;Hgb}{dL\;\;blood}\right)\times Hct.$$

As an input to Equation (6), we estimated *SvO*_2_ based on FD-DOS measurement of *StO*_2_ and arterial blood gas measurement of *SaO*_2_. Specifically, assuming a fixed venous fraction (γ) of blood monitored by FD-DOS, i.e., ${StO}_{2}={\left(1-\gamma \right)SaO}_{2}+{\gamma SvO}_{2}$ [49,50,51], we can estimate the cerebral venous blood oxygen saturation as:(7)$${SvO}_{2}=\frac{{StO}_{2}-(1-\gamma )\times {SaO}_{2}}{\gamma }$$

We assumed γ=0.75 to derive *SvO*_2_ and then substituted *SvO*_2_, along with the arterial blood gas measurement of *Hct*, into Equation (6) to estimate *CvO*_2_ for the *OEF* measurement.

#### 2.3.3. Cerebral Blood Flow (CBF)

The DCS-measured blood flow index (*BFI*) is proportional to cerebral blood flow (*CBF*) [38]. However, changes in hematocrit can significantly modify the proportionality between *BFI* and *CBF* [52,53]. To date, studies have largely assumed this proportionality remains constant when examining relative changes in *CBF* (i.e., they have assumed $rCBF=BFI/{BFI}_{baseline}$). In the present work, we used results from a previously derived shear-induced diffusion model of red blood cell motion [53] as a starting point to derive a modification to the proportionality between *BFI* and *CBF* (see Appendix A):(8)$$CBF=\frac{3\;{vol}_{RBC}\times {\;\mu }_{s,avg}^{\prime }\times R}{{8\;\alpha }_{shear}\times \sigma_{RBC}\times (1-g)}\times \frac{1}{Hct}\times BFI.$$

Here ${vol}_{RBC}$ is the average volume of a single red blood cell; $\mu_{s,avg}^{\prime }$ is the reduced scattering coefficient of tissue at the DCS light wavelength (which we measured concurrently with FD-DOS); R is the average radius of the blood vessels probed; $\alpha_{shear}$ is the proportionality between the shear flow rate and the red blood cell diffusion coefficient (empirically observed to be in the range of 10^−7^–10^−6^ mm^2^ [54,55]); $\sigma_{RBC}$ and g are the light scattering cross-section and light scattering anisotropy factor of a red blood cell at the DCS light wavelength; and Hct is the hematocrit. In the present work, relative *CBF* compared to baseline (i.e., $rCBF\equiv CBF/CBF_{0}$; $CBF_{0}$ denotes the mean *CBF* across the baseline period) were derived from DCS measurements of *BFI*, FD-DOS measurements of $\mu_{s,avg}^{\prime }$, and blood gas measurements of Hct using Equation (9). Assuming that ${vol}_{RBC}$, $\alpha_{shear}$, $\sigma_{RBC}$, g, and R all remained constant over time, rCBF was explicitly computed as (subscript 0 denotes mean baseline value):(9)$$rCBF=\frac{BFI\times \mu_{s,avg}^{\prime }}{BFI_{0}\times \mu_{s,avg,0}^{\prime }}\times \frac{Hct_{0}}{Hct}\times 100\%.$$

### 2.4. Statistical Analysis

Statistical analyses were carried out using MATLAB 2022a (The MathWorks Inc., Natick, MA, USA). All statistical tests were two-sided, and a *p*-value < 0.05 was used to deem significance. First, we assessed the average impact of mild hypothermic CPB on MAP, arterial blood gas metrics, cerebral optical metrics, and cerebral microdialysis biomarkers. For each animal in the cohort and for each measurement parameter, we computed the mean across the baseline period (i.e., the 5 min window just prior to CPB cannulation) and the mean across the mild hypothermic CPB period (i.e., the 140 window of constant temperature and CPB flow rate). Wilcoxon signed-rank tests were then used to assess differences between the baseline and CPB periods. Summary statistics for this analysis are presented using medians and interquartile ranges.

Next, we used linear mixed-effects regression analyses to investigate the presence of linear temporal trends in MAP, *Hct*, *CaO*_2_, cerebral optical metrics, and cerebral microdialysis biomarkers for each experimental period after baseline (i.e., CPB, hyperacute survival, acute survival). Prior to performing the regressions, MAP, *Hct*, *CaO*_2_, and cerebral optical metrics were first block-averaged across time windows that corresponded to each microdialysis sampling period (i.e., across 20 min windows prior to microdialysis collection during CPB, and across 60 min windows prior to microdialysis collection during survival; see Figure 1). Regressions were then performed on the relative changes of each parameter from baseline. The relative changes were computed as:(10)$$\Delta rParameter=\left(\frac{Parameter}{Parameter_{0}}-1\right)\times 100\%.$$

Recall, the subscript “0” denotes the mean across the baseline period. The linear mixed-effects analyses of temporal trends include random slope and intercept effects to account for within-subject correlation of repeated measures. The explicit time periods for the analyses were (a) mild hypothermic CPB (20–160 min CPB); (b) hyper-acute survival (0–8 h after decannulation); and (c) acute survival (8–24 h after decannulation). The cooling (0–20 min CPB) and rewarming (160–180 min CPB) periods were excluded due to the non-linear effects of temperature change. Note that we sought to differentiate the hyperacute and acute survival periods based on the hypothesis that the linear trends are different immediately after decannulation than at a longer time after decannulation. The eight-hour post-decannulation timepoint that we used to divide the hyperacute and acute survival periods was selected based on previous literature highlighting a period of vulnerability 8 to 12 h after decannulation [56,57,58].

For each linear model, the *p*-value of the y-intercept tests whether the percentage change from baseline significantly differs from 0 at the start of each experimental period, and the *p*-value of the slope tests whether there is a significant change in the parameter during the experimental period. Summary statistics for the fitted slope and intercept parameters are presented as the mean and 95% confidence interval.

Finally, of particular interest to us was examining the impact of correcting the optical *CBF* measurement for changes in optical scattering and hematocrit ($\mu_{s,avg}^{\prime }$ and *Hct*, respectively, in Equation (8)). Using additional linear mixed-effects regression analyses, we examined the impact of *CBF* correction (Equation (8)) on initial changes and temporal trends in *CBF* and *CMRO*_2_ during mild hypothermic CPB. In a supplementary analysis (see Appendix A) we also examined the comparative impact of correcting for free dissolved arterial oxygen (*PaO*_2_ in Equation (4)) and hematocrit (*Hct* in Equations (4) and (6)) in the calculation of *CaO*_2_ and *OEF*. Changes from baseline in correction parameters were most significant during CPB, and thus our analysis focused on this time period.

## 3. Results

Twenty-seven piglets (3.5–5.5 kg) were successfully placed on CPB for 3 h and maintained at 34 °C for 140 min. Eight animals were monitored for the CPB period only. In the remaining 19/27 animals, CPB support was discontinued and animals were monitored for either eight (*n* = 5), twelve (*n* = 5), eighteen (*n* = 4) or twenty-four (*n* = 5) hours after decannulation.

Continuous FD-DOS and DCS measurements of wavelength-dependent tissue optical absorption and scattering coefficients and blood flow index, respectively, were successfully collected throughout CPB in all animals. Overall, an average of 7% of DCS data were excluded from the CPB period and about 4% from the survival period due to the β fluctuations discussed in the Methods section. One animal did not have blood gas analysis due to equipment malfunction, limiting the possibility for hematocrit-corrected optical measurements; therefore, no blood gas data, nor optical properties requiring blood gas data, are included in the analysis for this animal. All optical data, unless otherwise described, reflect values corrected for dissolved oxygen, optical scattering, and hematocrit.

One animal did not have MD collection for the first hour of the experiment due to battery failure in the MD pump. Additionally, MD data were not included if they fell above or below detection limits. During the bypass period, 78% (*n* = 173) of glucose values, 83% (*n* = 183) of glycerol values, and 78% (*n* = 172) of LPR values were included, and during the survival period, 75% (*n* = 208) of glucose values, 95% (*n* = 262) of glycerol values and 91% (*n* = 250) of LPR values were included.

Summary statistics are reported as either median followed by interquartile range (IQR) in square brackets (i.e., median [IQR]), or as mean followed by 95% confidence interval in parentheses (i.e., mean (95% CI)). All parameter values reported in text reflect the relative change in the parameter as a percentage of baseline, i.e., Equation (10).

### 3.1. Cardiopulmonary Bypass (CPB)

The means of each parameter across the baseline period and across the mild hypothermic CPB period were compared. Summary results and statistics are listed in Table 1; we observed a significant increase in hematocrit (median [IQR] = +58 [26, 74] %, *p* < 0.001) during CPB which reflected the successful maintenance of hematocrit goals (*Hct* > 28%). Increased hematocrit, as well as a modest increase in *SaO*_2_ (+2.4 [2.0, 3.1] %, *p* < 0.001), resulted in a substantial increase in *CaO*_2_ (+65 [32, 80] %, *p* < 0.001). In addition, decreased *OEF* (−12 [−17, −1.9] %, *p* < 0.001) and *CBF* (−37 [−59, −11] %, *p* = 0.03), and increased *StO*_2_ (+13 [5.2, 22] %, *p* < 0.001), were observed. These latter changes compensated for the increase in *CaO*_2_ such that there was no significant change in *CMRO*_2_ from baseline during CPB (*p* = 0.8). There were also no changes in the microdialysis injury biomarkers, nor in optical scattering parameters measured by FD-DOS.

We next looked at temporal trends in measured systemic and cerebral physiologic parameters during CPB (Table 2). *OEF* was significantly reduced from baseline at the start of mild hypothermic CPB (mean (95% CI) = −12 (−16, −8) %, *p* < 0.001), and increased with time (+2.2 (0.0, 4.4) %/h, *p* < 0.001). This finding was also mirrored by *CBV*, i.e., *CBV* decreased at the start of mild hypothermic CPB (−21 (−26, −16) %, *p* < 0.001) and then increased with time (+2.0 (0.4, 3.6) %/h, *p* = 0.011). Interestingly, trends in *CBF* did not mirror *CBV*; *CBF* also initially decreased (−22 (−42, −2) %, *p* = 0.03), but then decreased with time (−5.5 (−10.5, −0.5) %/h, *p* = 0.03). The initial *CMRO*_2_ at the start of mild hypothermic CPB did not differ from baseline, but decreased with CPB time (−6.7 (−12.2, −1.2) %/h, *p* = 0.02). Of note, the *CMRO*_2_ decrease with time was not sufficient to make the mean *CMRO*_2_ significantly different from baseline in the Table 1 analysis. These results demonstrate that CPB has both an acute and sustained impact on cerebral physiology. No significant changes or trends in cerebral microdialysis biomarkers were observed.

The importance of correcting for hematocrit in the estimation of *CBF* and, dependently, *CMRO*_2_, is highlighted by the comparison of linear mixed-effects models with versus without hematocrit correction (Figure 2). At the start of mild hypothermic CPB, uncorrected *CMRO*_2_ demonstrated a significant increase from baseline (+58 (27, 90) %). This was a result of the combination of uncorrected *CBF*, which did not differ from baseline, with significantly increased *CaO*_2_ (+60 (48, 72) %), which predominated over the effect of a moderate decrease in *OEF* (−12 (−16, −8) %). However, hematocrit correction resulted in a significantly lower *CBF* than baseline and a *CMRO*_2_ that did not differ from baseline at the start of mild hypothermic CPB (−22 (−42, −2) %), as reported in Table 1 and Table 2. Hematocrit correction did not impact the direction of significant temporal trends. Both corrected and uncorrected *CBF* and *CMRO*_2_ demonstrated a significant decrease with increasing CPB time.

Results of the supplementary analysis comparing the impact of *PaO*_2_, *Hct*, and $\mu_{s,avg}^{\prime }$ correction on *CaO*_2_, *OEF*, *rCBF* and *rCMRO*_2_ estimation error during mild hypothermic CPB are included in Appendix A. This analysis underscores that among correction parameters, hematocrit had the greatest impact on error, specifically on *CaO*_2_ and *CBF*, which scale proportionally and inversely proportionally, respectively, with hematocrit. The errors attributed to neglecting changes in hematocrit were greatest in *CBF*, resulting in a median [IQR] = +58 [26, 72] % overestimation, followed by *CaO*_2_, resulting in a relative *CaO*_2_ underestimation error of −37 [−44, −22] %. Neglecting *PaO*_2_ resulted in a −2.8 [−3.0, −2.3] % error in *CaO*_2_ at baseline, which increased to a −4.9 [−5.2, −4.6] % error during mild hypothermic CPB. Similarly, neglecting *PaO*_2_ resulted in a −1.7 [−2.1, −1.3] % error in *OEF* at baseline, which increased to an error of −4.2 [−5.0, −3.6] % during mild hypothermic CPB. This increase in error is expected due to the hyperoxemia-associated increase in *PaO*_2_ during CPB. Finally, in line with our linear regression analysis, we did not observe a significant impact of $\mu_{s,avg}^{\prime }$ correction on *CBF* error. This result agrees with the absence of changes in $\mu_{s,avg}^{\prime }$ during and after CPB.

### 3.2. Hyper-Acute and Acute Survival Period

In the animals that were monitored after cessation of CPB support (i.e., after decannulation), the trends in measured systemic and cerebral physiologic parameters were analyzed from 0 to 8 h after decannulation, and from 8 to 24 h after decannulation. Summary results of the piecewise linear mixed-effects models for the two periods are listed in Table 3. A significant elevation in glycerol was observed at the beginning of the “hyper-acute” survival period (0–8 h after CPB; +48 (7, 90) %, *p* = 0.02), but glycerol levels returned to baseline in the later “acute” survival period (8–24 h after CPB). No significant difference in LPR values from baseline, nor trends over time were observed in either the hyper-acute or acute survival periods.

The increases observed during CPB in both hematocrit and *CaO*_2_ were sustained throughout the hyper-acute and acute survival periods; both parameters remained significantly higher than baseline (+50–60%). However, like glycerol, diffuse optical parameters demonstrated disparate trends between hyper-acute versus acute survival periods. Specifically, trends during the hyper-acute period matched the trends during CPB discussed above, i.e., *OEF* increased while *CBF* and *CMRO*_2_ decreased over time. In the subsequent acute survival period, temporal changes were not significant for any parameter except *CBF*, which shifted from a decrease over time in the hyper-acute survival period to an increase over time in the acute survival period. The alignment of hemodynamic and metabolic trends with glycerol (an injury biomarker) suggests a mechanistic role of these physiologic parameters in both the generation and resolution of injury.

## 4. Discussion

In this study, we characterized the effects of cardiopulmonary bypass on cerebral physiological parameters quantified non-invasively by diffuse optics and invasively by cerebral microdialysis in a neonatal swine model for up to 24 h following CPB support. An important advance of our optical study was the implementation of corrections for free dissolved arterial oxygen, optical scattering, and hematocrit in the optical quantification of cerebral blood flow and oxygen metabolism. We observed that hematocrit corrections were particularly important due to large hematocrit changes at the initiation of bypass. Another important finding of the study was alignment of transient temporal trends in optically measured cerebral metabolic parameters with transient elevation in cerebral glycerol, a biomarker of injury, within the first eight hours following CPB. Prior to elevation of glycerol, we observed significant temporal trends in cerebral metabolic parameters during CPB that persisted, concurrently with elevated glycerol, for eight hours following CPB. In the 8–24 h timescale, the glycerol elevation resolved and prior significant temporal trends in cerebral metabolic parameters were no longer present. Our results suggest that during this hyper-acute period (0–8 h after CPB) the brain may be at increased vulnerability to secondary insults. 

### 4.1. Improved Optical Quantification of CMRO_2_: Impact of Hematocrit

In many clinical settings, substantial changes in hematocrit occur. These changes can influence interpretations of diffuse optical *CBF* and *CMRO*_2_ measurements. Our CPB swine model offered an opportunity to critically explore this issue. We observed a significant increase in hematocrit during mild hypothermic CPB (median [IQR] = +58 [26, 74] % relative to baseline), which remained high throughout the entire 24 h survival window. The increase was anticipated because baseline hematocrit (23 [20, 27] %) was consistently lower than the target hematocrit goal during CPB. Prime blood within the CPB circuit was consistently hemoconcentrated to elevate hematocrit levels to achieve an equilibrated hematocrit during CPB support of >28%. Note that in addition to CPB, hemoconcentration occurs in other clinical situations including extracorporeal membrane oxygenation (another form of life support) and blood transfusions. Hematocrit levels can also be reduced when there is severe blood loss or when an excess of fluids is given to treat hypotension or severe dehydration [59,60]. In all of these scenarios, we believe that optical *CBF* and *CMRO*_2_ metrics should be adjusted to account for hematocrit variation. Proper accounting aids in accurate characterization of longitudinal changes and in comparisons across subjects with different hematocrit levels.

Specifically, employing the conventional DCS approach that does not account for hematocrit changes from baseline (i.e., CBF∝BFI), we observed an essentially negligible (2%) increase in *CBF* from baseline at the beginning of MH-CPB. We then explored the use of different hematocrit correction formulations from recent research. Recently, the effect of hematocrit on *CBF* quantification has been examined in an in vitro experimental study and in theory/simulation work; the two studies gave conflicting results [52,53]. In the first scheme, when effects of increasing hematocrit were examined in vitro in microfluidic phantoms, the DCS-measured *CBF* was found to decrease with increasing red blood cell concentration; specifically: CBF ∝ BFI/(1−1.8×Hct) [52]. Using this approach to correct our data resulted in a mean (95% CI) = 74 (39, 110) % increase in *CBF* from baseline at the start of MH-CPB (as well as a 148 (97, 200) % increase in *CMRO*_2_ from baseline). For our experimental swine model, this sudden and sustained increase in *CBF* is non-physiologic. Such a large change would produce hyperemic injuries (edema and/or hemorrhage) that were not observed [61,62,63], and it is also difficult to explain such a large discontinuous change in oxygen metabolism at the onset of MH-CPB.

We also explored a second approach to account for the effects of hematocrit, which seemed to work better for our experiment. Specifically, we developed a correction based on the theory/simulation model of Boas et al. [53]. For the same *BFI*, this correction gives a decrease in *CBF* with increasing *Hct*: CBF ∝ BFI/Hct (refer to Appendix A). We applied this approach to our swine data, and we found a plausible decrease in *CBF* at the onset of MH-CPB of −22 (−42, −2) %. This level of reduction in *CBF* with reduction in temperature has been reported during hypothermic CPB [23,29], as well as in the setting of hyperoxemia [64]. Our data, based on the inverse relationship between *CBF* and *Hct*, are thus consistent with physiologic expectations.

In addition to accounting for the influence of hematocrit on *CBF* estimation, we also accounted for the influence of hematocrit and free dissolved arterial oxygen content in our calculation of *CaO*_2_ and *OEF* to more accurately estimate *CMRO*_2_. *CaO*_2_ increases proportionally with hematocrit. Neglect of large hematocrit changes induced by CPB-associated hemoconcentration resulted in a proportional underestimation of *CaO*_2_; impact on *OEF* was marginal (<5%). When no correction was applied, large errors resulted in *CaO*_2_ and *rCBF*; yet we observed relatively smaller errors in *rCMRO*_2_ (see Appendix A). This result follows from the cancellation of the proportional hematocrit correction of *CaO*_2_ with the inversely proportional hematocrit correction of *CBF*; however, a ~10% error remained due to neglect of free dissolved arterial oxygen. Altogether, these findings underscore the outsized influence of hematocrit correction on accurate physiologic estimation of *CaO*_2_, *CBF*, and *CMRO*_2_. Importantly, when we implemented novel *Hct* correction of *CBF* in vivo, the *CMRO*_2_ values at the start of and during CPB did not significantly differ from baseline. Future work is needed to understand the discrepancy of our approach with respect to the results of the in vitro measurements. For example, some of the parameters in the relation between *CBF* and *BFI* (Equation (8)) which we have assumed to be constant, such as $\alpha_{shear}$ and ${vol}_{RBC}$, could depend on hematocrit. Additionally, further validation of these corrections against gold standard measurements of *CBF* and *CMRO*_2_ is important.

### 4.2. Effects of Cardiopulmonary Bypass

As discussed above, we did not observe a significant reduction in *CMRO*_2_ from baseline during mild hypothermic CPB (i.e., assuming CBF∝BFI/Hct in our analysis). This finding contrasts with prior observations of reduced metabolic demand during hypothermia [65,66,67]; however, thermoregulatory release of catecholamines leading to increased vascular tone may account for compensatory maintenance of *CMRO*_2_ at mild hypothermic temperatures [68,69]. The lack of reduction in *CMRO*_2_ could indicate a need to cool further than 34 °C to reduce tissue metabolism as a means of end-organ protection [68].

Significant changes were observed in other associated parameters. Specifically, due to the increase in hematocrit, *CaO*_2_ increased from baseline, and *OEF*, *CBF*, and *CBV* concurrently decreased. These findings agree with previous observations of an acute vasoconstrictive response to isovolemic hemoconcentration, whereby a 20% increase in *Hct* resulted in a 10% decrease in *CBF* and a 4.4% decrease in basilar artery diameter [70]. Thus, in the context of the 53 (42, 63) % increase in hematocrit at the start of mild hypothermic CPB, our findings of a 22 (2, 42) % decrease in *CBF* and 21 (16, 26) % decrease in *CBV* are plausible. Moreover, since the decrease in *CBF* does not completely compensate for the increase in *CaO*_2_, a decrease in *OEF* is expected when *CMRO*_2_ is maintained.

During CPB, we observed a decreasing trend in *CBF* and *CMRO*_2_ and an increasing trend in *OEF* and *CBV*. Increasing *OEF* indicates a growing imbalance between oxygen delivery and demand. The lack of elevation of biomarkers of cerebral metabolic distress and injury suggests that, despite growing mismatch, for the three-hour-duration experiment, oxygen demands were met. Note that for longer CPB times, if the observed *CBF* and *OEF* trends persist, then it is possible that oxygen demands may not be met. The discrepant direction of *CBF* and *CBV* trends are also notable and point to a potential inflammatory mechanism which may elevate intracranial pressure and modify cerebral perfusion pressure independently of vasomotor tone.

Recent work has highlighted the role of inflammation, mitochondrial dysfunction, and reactive oxygen species generation as mechanisms of neurological injury during MH-CPB [8]. A decreasing *CBF* could result from an associated increase in cerebral edema leading to increasing intracranial pressure and reducing cerebral perfusion pressure [71,72]. Increasing trends in *OEF* and *CBV* suggest a compensatory metabolic and vasomotor response to the decreasing *CBF*. Alternatively, the increases in *OEF* and *CBV* could be responses to increasing mitochondrial dysfunction (e.g., increasing leakiness of inner mitochondrial membrane leading to decreased energy production per molecule of oxygen) to meet metabolic demands. Further study is needed to elucidate whether the observed *CMRO*_2_ trends reflect a primary decrease in metabolic demand or, alternatively, a reduction in mitochondrial efficiency of oxygen utilization.

### 4.3. Cerebral Metabolism and Injury during Post-CPB Survival (0–24 h)

In the first eight hours following CPB (termed “hyper-acute” survival in our work), we observed an elevation in cerebral glycerol, a biomarker of injury. While this injury was observed following CPB, related trends in cerebral physiologic parameters suggest that this injury process was initiated during CPB and continued for eight hours following decannulation. During this period, consistently with temporal trends observed during CPB, *OEF* significantly increased, and *CBF* and *CMRO*_2_ significantly decreased over time. In the subsequent 8–24 h “acute” survival period, glycerol was no longer elevated, and these cerebral physiologic trends were no longer present. In fact, *CBF* demonstrated an inflection and began to increase over time. These results suggest that CPB-associated injury mechanisms persisted for eight hours following CPB and thus point to this period as a critical window of neurological vulnerability. Further, the common hemodynamic pattern observed, i.e., the reduction in *CBF* and *CMRO*_2_ with corresponding increase in *OEF*, may serve as a biomarker of this injury process. 

In a previous study, applying diffuse optical quantification to characterize the first 12 h following CPB in neonates with severe congenital heart defects who underwent either normothermic or mild hypothermic (>28 °C) CPB, a reduced *CBF* was also observed following CPB compared to preoperative values [32]. Clinical subjects also exhibited decreased *OEF* after decannulation. However, in our preclinical subjects, *OEF* did not significantly differ from baseline values immediately after decannulation, then decreased with time during the hyper-acute period, and stabilized but remained moderately elevated (+7 (2, 12) % from baseline) during the acute survival period from 8 to 24 h after CPB. Since our model used healthy animal subjects to isolate the effects of CPB, the differences in *OEF* across studies may be a result of impaired oxygenation at baseline in clinical subjects which was corrected during surgery.

CPB is known to result in systemic inflammation; recent work by Tu et al. highlights the predominant role of blood shear stress in upregulating transcription of proinflammatory cytokines (IL-8 and TNF-α) which disrupt the integrity of the blood–brain barrier [57]. Pathologic extravasation of ions, inflammatory mediators, and large molecules leads to progressive tissue inflammation/edema and cerebral metabolic dysfunction (e.g., mitochondrial dysfunction) ending in necrosis and apoptosis. In healthy 1-month-old swine who underwent 2 h of mild hypothermic (30 °C) CPB, expression of IL-8 and TNF-α were observed to increase with CPB time, then remain elevated at 1 h after CPB, and finally resolve by 6 h [57]. The inflection of physiologic trends between the hyper-acute and acute survival periods that we observed are consistent with this and other prior evidence of the timing of an inflammatory response [73,74]. Concurrent MAP elevation with diminished *CBF* during the survival period lends further support to the hypothesized mechanism of increased intracranial pressure that may occur secondary to inflammation [75,76]. The use of optical neuromonitoring may aid in monitoring the efficacy of therapeutic strategies to reduce inflammation, and subsequent injury, during and following CPB. Advanced optical techniques to non-invasively estimate intracranial pressure are under development and could provide further discrimination in the future [77,78,79,80,81,82].

### 4.4. Limitations

To calculate *OEF* changes, we assumed the venous fraction (γ) of blood monitored by FD-DOS remained constant over time. γ has been reported to increase with hyperoxemia due to arteriolar vasoconstriction [83]. However, within the physiologic ranges of our experimental model, the γ increase due to hyperoxemia is expected to be less than 5%. Use of the constant-γ assumption potentially resulted in underestimation of *OEF* and *CMRO*_2_ proportional to the relative increase in γ. We also do not expect CPB to substantially alter γ, but future work using invasive cerebral venous sampling is needed to confirm this expectation. For *CBF* measurements, we also assumed that the average blood vessel radius in the DCS-sampled tissue volume, the average individual red blood cell volume, the average red blood cell’s scattering cross-section and scattering anisotropy coefficient, and the proportionality between shear rate and red blood cell diffusion all remained constant over time. The validity of these assumptions during/after CPB needs to be confirmed in future work against gold standard *CBF* measurements.

Our study results are also limited by the duration of monitoring after CPB (up to 24 h after CPB); secondary injuries that have been observed later in the recovery period may be missed [11,32,84]. Neuroimaging, which is commonly used as a clinical marker of neurological injury [22,85], was not performed either before or after CPB. Additionally, only averages and linear regression analysis were used to identify significant changes in physiologic parameters from baseline. Transient inflections during CPB or during the survival period may not be captured by our results.

The use of healthy swine may also impact the translation of results to neonatal patients. First, the piglet baseline temperature is slightly higher at 38.5 °C than human neonates at 36.5 °C [23,86]. Additionally, the oxygen binding affinity of swine hemoglobin is lower than human hemoglobin and could account for comparatively lower baseline oxygen saturations [87]. It is also important to note that baseline hematocrit in healthy human neonates is around 57% [88]. Although hematocrit is lower in neonates at the beginning of reparative cardiac surgery (around 35% [89]), it still exceeds baseline hematocrit levels in our swine cohort (around 25%). Thus, hematocrit corrections during CPB are expected to have different trends and magnitudes for infants. Also, the relatively robust hemodynamics in our healthy swine neonates (compared to critically ill, peri-operative human infants) led to routine administration of vasodilators to maintain MAP targets in this study. Thus, the infrequent use of vasodilators in clinical practice may also contribute to discrepancies between our results and observed clinical trends. Another limitation of our study is the use of only female swine, which reduces heterogeneity in these pilot studies. Since there are reported differences between piglet sex per brain development and neurological injury, further studies are required to understand potential sex differences [90,91,92,93].

## 5. Conclusions

This work features the novel in vivo application of diffuse optical approaches which account for hematocrit during *CBF* and *CMRO*_2_ quantification in a neonatal swine model of cardiopulmonary bypass and survival. Our results highlight that the application of hematocrit correction was critical to achieving physiologically plausible changes in *CBF* and *CMRO*_2_. Using hematocrit correction, optical monitoring showed temporal decreases in cerebral blood flow with concurrent increases in cerebral oxygen extraction and cerebral blood volume fraction during CPB and in the hyper-acute 8 h period after the end of CPB. Concurrent cerebral microdialysis measurements of glycerol were also elevated during the 8 h hyper-acute period. In combination, these results suggest a possible inflammatory mechanism of injury from CPB. The return of glycerol to baseline and the reversal of the cerebral blood flow and oxygen extraction fraction trends during the 8–24 h period after CPB suggests the CPB period and the hyperacute post-CPB phase are important windows of neurological vulnerability. In the future, non-invasive diffuse optical neuromonitoring of injury processes associated with CPB may aid in the optimization of clinical management during periods of increased neurological vulnerability.

## Figures and Tables

**Figure 1 metabolites-13-01153-f001:**
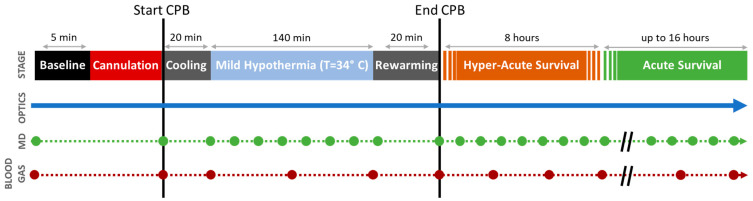
Protocol Timeline. Continuous diffuse optical neuromonitoring was performed concurrently with episodic cerebral microdialysis sampling (MD, green circles) and arterial blood gas sampling (BG, red circles) before/during/after cardiopulmonary bypass (CPB). Optical statistical analysis was performed using three linear mixed-effects models looking at three distinct periods: mild hypothermic CPB, hyper-acute survival, and acute survival. The intercepts from these models show the initial changes from baseline (i.e., acute effects) for each period; the slopes indicate trends during each period.

**Figure 2 metabolites-13-01153-f002:**
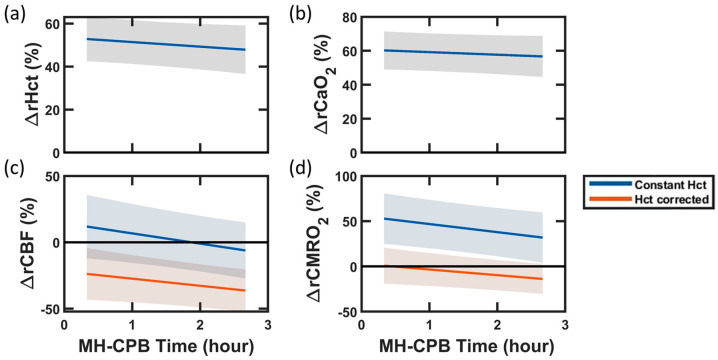
Impact of Hematocrit Correction on Blood Flow and Oxygen Metabolism During Mild Hypothermic Cardiopulmonary Bypass (MH-CPB). Linear mixed-effects regression analyses of percent changes from baseline in (**a**) hematocrit (*Hct*), (**b**) arterial oxygen content (*CaO*_2_), (**c**) cerebral blood flow (*CBF*), and (**d**) cerebral metabolic rate of oxygen (*CMRO*_2_) during the 140 min of MH-CPB. The linear fit (solid lines) and 95% CI (shaded regions) across the 27 swine are shown in all plots. In panel (**c**), *CBF* was derived in two ways: (1) from DCS measurements assuming constant *Hct* (blue line), and (2) from DCS measurements, blood gas *Hct* measurements, and FD-DOS scattering measurements using Equation 8 (red line). In panel (**d**), *CMRO*_2_ was derived from Equation (3) using (1) the “constant Hct” *CBF* measurements (blue line), and (2) the “Hct corrected” *CBF* measurements (red line).

**Table 1 metabolites-13-01153-t001:** Summary Statistics of Baseline and Bypass Periods.

Modality	Parameter	Baseline	Mild Hypothermic CPB	*p*-Value
Animal Characteristics	Length (cm)	37 [36, 38], (*n* = 27)	-	-
	Weight (kg)	4.3 [4.1, 4.6], (*n* = 27)	-	-
Vitals	MAP (mmHg)	61 [53, 72], (*n* = 27)	64 [57, 71], (*n* = 27)	0.3
Blood Gas	Hct (%)	23 [20, 27], (*n* = 27)	34 [33, 35], (*n* = 27)	**<0.001**
	CaO_2_ (mL O_2_/dL blood)	10 [9, 12], (*n* = 26)	16 [15, 16], (*n* = 26)	**<0.001**
	SaO_2_ (%)	98 [97, 98], (*n* = 26)	100 [99.9, 100], (*n* = 26)	**<0.001**
	PaO_2_ (mmHg)	91 [84, 96], (*n* = 26)	257 [237, 271], (*n* = 27)	**<0.001**
	PaCO_2_ (mmHg)	39 [37, 41], (*n* = 26)	45 [42, 48], (*n* = 26)	**<0.001**
	Glucose (mg/dL)	102 [79, 117], (*n* = 27)	150 [120, 184], (*n* = 27)	**<0.001**
	Lactate (mmol/L)	1.2 [1, 1.6], (*n* = 27)	2.2 [1.6, 2.9], (*n* = 27)	**<0.001**
Optics: Physiologic Parameters	StO_2_ (%)	54 [50, 57]_,_ (*n* = 27)	61 [58, 64], (*n* = 27)	**<0.001**
	OEF (%)	62 [56, 66]_,_ (*n* = 26)	55 [50, 58], (*n* = 26)	**0.002**
	rCBF (% Baseline)	100, (*n* = 27)	63 [41, 89], (*n* = 27)	**0.004**
	rCMRO_2_ (% Baseline)	100, (*n* = 26)	91 [57, 135], (*n* = 26)	0.5
	CBV (μL/g brain tissue)	53 [49, 67], (*n* = 27)	48 [38, 51], (*n* = 27)	**<0.001**
	THC (μM)	68 [61, 82]_,_ (*n* = 27)	82 [74, 89], (*n* = 27)	**<0.001**
Optics: Mie Scattering Parameters	A	14 [12, 17], (*n* = 27)	14 [10, 16], (*n* = 27)	0.6
	b	1.1 [0.95, 1.3], (*n* = 27)	0.98 [0.80, 1.23], (*n* = 27)	0.2
	μ_s_′(785 nm)	8.3 [7.8, 9.7], (*n* = 27)	8.3 [7.1, 9.8], (*n* = 27)	0.8
Microdialysis	LPR	17 [15, 20], (*n* = 20)	15 [12, 20], (*n* = 25)	0.12
	Lactate (mM)	0.82 [0.63, 0.99], (*n* = 19)	0.65 [0.45, 0.86], (*n* = 25)	0.3
	Pyruvate (μM)	46 [33, 58], (*n* = 22)	35 [28, 54], (*n* = 27)	0.3
	Glycerol (μM)	21 [17, 31], (*n* = 21)	22 [17, 24], (*n* = 27)	0.7
	Glucose (μM)	12 [8, 19], (*n* = 20)	11 [6, 15], (*n* = 26)	0.17

Summary parameters are reported as median [IQR]. Reported *p*-values are associated with Wilcoxon Sign-Rank tests between Baseline and MH-CPB period for each parameter. Significant *p*-values are bolded for clarity. MAP = mean arterial pressure, Hct = hematocrit, CaO_2_ = arterial oxygen content, SaO_2_ = arterial blood-oxygen saturation, PaO_2_ = partial pressure of oxygen in the arterial blood, PaCO_2_ = partial pressure of carbon dioxide in the arterial blood, StO_2_ = tissue oxygen saturation, OEF = oxygen extraction fraction, rCBF = relative cerebral blood flow, rCMRO_2_ = relative cerebral metabolic rate of oxygen, CBV = cerebral blood volume, THC = total hemoglobin concentration. A and b are the Mie scattering parameters (i.e., $\mu_{s}^{\prime }\left(\lambda \right)=A{\left(\lambda /(500\;nm)\right)}^{-b}$ and $\mu_{s}^{\prime }(785\;nm)$ is the tissue reduced scattering coefficient at the 785 nm wavelength. LPR = lactate–pyruvate ratio.

**Table 2 metabolites-13-01153-t002:** Trends During the Mild Hypothermic Cardiopulmonary Bypass (MH-CPB) Period.

Parameter	LME Y-Intercept			LME Slope		
	Value (%)	95% CI	*p*-Value	Value (%/h)	95% CI	*p*-Value
ΔrMAP	+8	(−2, 18)	0.1	−1.0	(−4.1, 2.2)	0.6
ΔrHCT	+53	(42, 63)	**<0.001**	−2.1	(−5.0, 0.9)	0.16
ΔrCaO_2_	+60	(48, 72)	**<0.001**	−1.4	(−4.4, 1.5)	0.3
ΔrStO_2_	+13	(7, 19)	**<0.001**	−1.6	(−3.4, 0.2)	0.09
ΔrOEF	−12	(−16, −8)	**<0.001**	+2.2	(0.0, 4.4)	**0.046**
ΔrCBF	−22	(−42, −2)	**0.03**	−5.5	(−10.5, −0.5)	**0.03**
ΔrCMRO_2_	+4	(−17, 25)	0.7	−6.7	(−12.2, −1.2)	**0.02**
ΔrCBV	−21	(−26, −16)	**<0.001**	+2.0	(0.4, 3.6)	**0.01**
ΔrTHC	+17	(13, 21)	**<0.001**	+0.7	(−0.6, 1.9)	0.3
ΔrA	−6	(−11, −1)	**0.017**	−1.3	(−3.0, 0.5)	0.2
Δrb	−8	(−17, 1)	0.1	−1.8	(−5.3, 1.7)	0.3
Δrμ_s_^’^(785 nm)	−1.8	(−5.0, 1.4)	0.3	0.2	(−5.8, 1.4)	0.7
ΔrLactate	+17	(−27, 62)	0.4	−1.7	(−20, 17)	0.9
ΔrPyruvate	−5	(−27, 18)	0.7	9.5	(−8, 27)	0.3
ΔrLPR	+48	(−42, 138)	0.3	−23	(−65, 18)	0.2
ΔrGlycerol	+11	(−205, 226)	0.9	+51	(−86, 187)	0.5
ΔrGlucose	+47	(−23, 122)	0.2	−18	(−57, 24)	0.4

Results of the univariate linear mixed-effects (LME) models during MH-CPB accounting for within-subject correlation of repeated measures. In the linear mixed-effects model, 95% CI is the 95% confidence interval for the stated fixed-effect coefficient. Significant *p*-values are bolded for clarity.

**Table 3 metabolites-13-01153-t003:** Trends During the Hyper-Acute (0–8 h) and Acute (8–24 h) Survival Periods.

Parameter	Time after Decannulation	LME Y-Intercept			LME Slope		
		Value (%)	95% CI	*p*-Value	Value (%/h)	95% CI	*p*-Value
ΔrMAP	0–8 h	+12	(−5, 29)	0.2	+0.6	(−1.2, 2.4)	0.5
	8–24 h	+13	(1, 24)	**0.03**	−0.0	(−0.1, −0.0)	**<0.001**
ΔrHct	0–8 h	+53	(50, 55)	**<0.001**	−0.3	(−1.2, 0.7)	0.6
	8–24 h	+51	(40, 62)	**<0.001**	−0.4	(−0.5, −0.2)	**<0.001**
ΔrCaO_2_	0–8 h	+60	(49, 71)	**<0.001**	−0.5	(−0.6, −0.4)	**<0.001**
	8–24 h	+57	(46, 68)	**<0.001**	−0.6	(−0.7, −0.5)	**<0.001**
ΔrStO_2_	0–8 h	+6	(2, 10)	**0.003**	−1.0	(−1.4, −0.6)	**<0.001**
	8–24 h	−2	(−6, 1)	0.2	−0.0	(−0.6, 0.6)	0.9
ΔrOEF	0–8 h	−2	(−6, 3)	0.5	+1.0	(0.6, 1.4)	**<0.001**
	8–24 h	+7	(2, 12)	**0.003**	−0.1	(−0.8, 0.6)	0.8
ΔrCBF	0–8 h	−28	(−42, −14)	**<0.001**	−2.1	(−3.6, −0.6)	**0.005**
	8–24 h	−44	(−56, −33)	**<0.001**	+0.7	(−0.0, 1.4)	**0.05**
ΔrCMRO_2_	0–8 h	+9	(−10, 28)	0.3	−3.0	(−5.2, −0.8)	**0.008**
	8–24 h	−12	(−27, 2)	0.09	+0.6	(−0.2, 1.4)	0.1
ΔrCBV	0–8 h	−20	(−28, −11)	**<0.001**	−1.0	(−1.5, −0.5)	**<0.001**
	8–24 h	−28	(−38, −18)	**<0.001**	+0.1	(−0.4, 0.6)	0.7
ΔrTHC	0–8 h	+18	(11, 26)	**<0.001**	−1.7	(−2.4, −1.1)	**<0.001**
	8–24 h	+4	(−6, 14)	0.5	+0.1	(−0.6, 0.8)	0.8
ΔrA	0–8 h	−6	(−13, 0.6)	0.08	+0.1	(−1.0, 1.2)	0.8
	8–24 h	−7	(−18, 5)	0.2	+0.7	(−1.3, 2.7)	0.5
Δrb	0–8 h	−6	(−23, 10)	0.4	+1.1	(−1.4, 3.6)	0.4
	8–24 h	−1	(−20, 18)	0.9	+1.1	(−2.5, 4.7)	0.6
Δrμ_s_^’^(785 nm)	0–8 h	−1	(−6, 3)	0.5	−0.3	(−1.3, 0.6)	0.5
	8–24 h	−5	(−15, 5)	0.3	+0.7	(−1.3, 2.7)	0.2
ΔrLactate	0–8 h	+127	(−152, 406)	0.4	+19	(−20, 57)	0.3
	8–24 h	+191	(−144, 526)	0.3	−0.2	(−4.3, 3.9)	0.9
ΔrPyruvate	0–8 h	+87	(−21, 195)	0.1	+1.5	(−2.2, 5.3)	0.4
	8–24 h	+97	(−33, 228)	0.1	−4.2	(−9.6, 1.1)	0.1
ΔrLPR	0–8 h	−5	(−40, 29)	0.8	+5.0	(−0.7, 10.7)	0.08
	8–24 h	+2	(−39, 43)	0.9	+4.0	(−11.7, 19.6)	0.6
ΔrGlycerol	0–8 h	+48	(7, 90)	**0.02**	+9.3	(−14.7, 33.3)	0.4
	8–24 h	+25	(−15, 64)	0.2	−2.0	(−8.0, 4.1)	0.5
ΔrGlucose	0–8 h	−0	(−94, 93)	>0.99	+2.1	(−21.6, 25.8)	0.9
	8–24 h	+60	(−101, 221)	0.5	+1.8	(−10.3, 14.0)	0.8

Results of the univariate linear mixed-effects models for optical parameters versus time after MH-CPB. The survival period was split into two phases, hyper-acute and acute survival, separated at eight hours after decannulation. Significant *p*-values are bolded for clarity.

## Data Availability

The data and code presented in this study are available on request from the corresponding author. The data are not publicly available due to privacy.

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
