# Peer review of "Diffuse Optical Monitoring of Cerebral Hemodynamics and Oxygen Metabolism during and after Cardiopulmonary Bypass: Hematocrit Correction and Neurological Vulnerability"

_metabolites, 2023, doi:10.3390/metabo13111153_

Round 1

Reviewer 1 Report

Comments and Suggestions for Authors

The manuscript is focused on effects of cardiopulmonary bypass (CPB) on CBF and CMRO2 as measured with diffuse optical methods. The authors should be congratulated on conducting such a large and impressive study. There are several outstanding contributions made by the manuscript. In particular, the authors show the influence of hematocrit on CBF and CMRO2 measurements, which has important implications on the DCS community. Overall the paper is well written, very comprehensive and detailed. There are only a few minor suggestions provided and clarifications needed.

1)      Abstract: “The hematocrit correction approach enabled detection of increased neurological vulnerability during and acutely following CPB” – this is technically not true. Neuro vulnerability is a bit more than accurate CBF and CMRO2 detection. Consider tuning down that statement.

2)      Explain the use of only female animals and if there are sex specific differences expected

3)      Please specify what the range of end-tidal CO2 was – not just target range.

4)      “To maintain as much homogeneity as possible across subjects prior to CPB, MAP was kept between 45-65 mmHg (i.e., via venous infusions of milrinone, nitroglycerin and/or nicardipine).” – did you record the venous infusion and how much did it vary between animals? Also, what is the effect of MAP being controlled on the data recorded and is the procedure comparable to CPB in humans?

5)      NIRS/DCS source detector position: was the probe location similar between animals? What is the distance to the brain from surface (to out source detector distance in context of distance to brain)?

6)      Beta<0.03 – can you please provide a reference for this value or explain how the value was determined for noise rejection?

7)      Assuming gamma to be constant: The authors discuss the point of assuming gamma to be constant in the discussion, but can you include an analysis of CMRO2 variation if gamma varied by the expected amount? How much variation is expected?

Reviewer 2 Report

Comments and Suggestions for Authors

The authors have investigated an important aspect in this area and have presented the results well. However, here are a few concerns that need to be addressed:

1. The introduction should include more examples from the available literature to make it further comprehensive.

2. The use of diffuse optical monitoring of cerebral hemodynamics in terms of hematocrit correction and neurological vulnerability has been well studied so far. Please highlight the novel edge of this study.

3. Try to remove the repetitive information in the introduction and conclusions.

Comments on the Quality of English Language

English is fine, try to do grammar formatting.
